# When pits fill up: Supply and demand for safe pit-emptying services in Kisumu, Kenya

**Rachel Peletz[1], Clara MacLeod[2], Joan Kones[2], Edinah Samuel[2], Alicea Easthope-Frazer[1], Caroline Delaire[1], Ranjiv Khush[1]***

**1** The Aquaya Institute, San Anselmo, CA, United States of America, **2** The Aquaya Institute, Nairobi, Kenya

* ranjiv@aquaya.org

## Abstract

Improving sanitation conditions in low-income communities is a major challenge for rapidly growing cities of the developing world. The expenses and logistical difficulties of extending sewerage infrastructure have focused increasing attention on the requirements for safe and cost-effective fecal sludge management services. These services, which are primarily provided by the private sector, include the collection and treatment of fecal waste from latrine pits and septic tanks. To determine the degree to which market forces can promote safe fecal sludge removal in low-income neighborhoods of Kisumu, Kenya, we compared household willingness-to-pay for formal pit emptying with the prices charged by service providers. Through surveys of 942 households and a real-money voucher trial with 646 households, we found that stated and revealed demand for formal emptying services were both low, with less than 20% of households willing to pay full market prices. Our results suggest that improving fecal sludge management in these neighborhoods via the private sector will require large subsides, ranging from 55.1–81.4 million KES (551,000–814,000 USD) annually, to address the gap between willingness-to-pay and market prices. Raising and administering subsidies of this scale will require the development of a city-wide sanitation master plan that includes investment, management, and regulatory procedures for fecal sludge management. In the absence of government investment and coordination, it is unlikely that the private sector will address safe sanitation needs in low-income areas of Kisumu.

## Introduction

Residents of low-income, urban settlements in developing countries generally do not have access to formal, regulated fecal sludge management (FSM) services, which primarily target middle- and high-income households that can afford market prices [1]. FSM in poor neighborhoods commonly includes unsafe practices, including the employment of informal manual emptiers who remove fecal sludge by hand and dispose of it in the surrounding environment. These practices pose environmental and health risks to households and emptiers, who often operate without proper equipment, training, or oversight [2–4]. As a result, inadequate sanitation service provision disproportionately affects the health and environment of the urban poor: a 2010 study of sanitation subsidies in six low-income countries found that that the

**Data Availability Statement:** The data are available at doi.org/10.17632/6w49x86k6y.1.

**Funding:** This research was delivered through a research agreement between Water & Sanitation

for the Urban Poor (WSUP) and The Aquaya Institute. The agreement was funded under the Urban Sanitation Research Initiative managed by WSUP and supported by UK Aid from the UK government.

**Competing interests:** The authors have declared that no competing interests exist.

majority of poor households pay high costs for onsite sanitation, an activity seen as a private good and a household responsibility, despite the negative public health externalities of poor sanitation [5].

Safe emptying services are inevitably more expensive than informal practices due to licensing requirements, the costs of transporting fecal sludge to designated disposal areas, and dumping fees [3]. Studies in different settings have found that that the willingness-to-pay (WTP) for safe latrine emptying services among low-income households is substantially lower than market prices: for example, in rural Bangladesh household contributions covered approximately 47% of service provider costs [6], and in urban Rwanda, affordable price points only covered approximately 40% of service provider costs for formal manual emptying [5]. Quantifying the differences between market prices (supply) and household WTP (demand), and, thereby, the extent to which low-income customers can bear the cost of safe emptying, is central to designing appropriate service delivery and financing approaches.

Various approaches, however, are emerging to subsidize sanitation for the poor, particularly in urban settings. In Dhaka, Bangladesh, for example, the SWEEP fecal waste collection service is implementing an internal cross-subsidy mechanism to serve the poor. Customers in low-income areas are charged less than middle/high-income customers, and, after achieving commercial viability in mid-2017, SWEEP must now ensure that 30% of customers are actually classified as low-income based on two parameters: their geographic location in a low-income area and confirmation of income status by customer [7]. Financial modeling indicates that a 70–30% mix of high-income/institutional and low-income customers will ensure sufficient profit to maintain private-sector interest in the service delivery partnership. Sanitation surcharges that are levied on top of utility customer charges provide another strategy for subsidizing the provision of sanitation services and infrastructure [8]. In Africa, at least six cities have implemented a sanitation surcharge, and in three of these cities (Lusaka, Zambia; Bobo-Dioulasso, Burkina Faso; and Ouagadougou, Burkina Faso) the resulting revenues are earmarked for pro-poor services [9]. As of 2010, the surcharge in Burkina Faso had supported improved sanitation infrastructure for nearly one million people, without additional donor inputs [10]. By 2012, the surcharge in Lusaka had raised at least $2 million, and disbursements had funded 200 onsite sanitation facilities and condominial sewerage in low-income peri-urban areas [9].

Other pro-poor sanitation improvement efforts address the costs of service provision. In Dakar, Senegal for example, the national sanitation agency ONAS (Office National de l'Assainissement du Sénégal) trained and certified private operators and created a call center to coordinate mechanized emptying services. The call center collects bids from vacuum truck operators for emptying jobs, which, reportedly has lowered prices by 20% and promoted increased use of vacuum trucks in low-income areas of the city [7]. Low-cost emptying technologies are also emerging: the Gulper, a manually operated latrine emptying pump designed for densely populated urban areas is now one of the most common manually-driven systems for removing fecal sludge in Africa; similarly, the Vacutug is a self-propelled mechanized vacuum pump successfully introduced in low-income areas of Nairobi, Kenya, and in Dar es Salaam, Tanzania [11,12].

Additional FSM innovations combine technology development with new service delivery models at household and community levels. For example, multiple organizations (such as SOIL in Haiti, Sanivation in Naivasha, Kenya, and Clean Team in Kumasi, Ghana) support the development and use of container-based sanitation (CBS) technologies for households. Container-based toilets collect fecal waste in a sealable, removable receptacle for subsequent transport to a treatment facility [13]. Sanergy has established a franchise system for community-level CBS in a low-income area of Nairobi, Kenya [14]. In these cases, treated and processed

fecal waste is sold (e.g., as fertilizer, an energy source, or animal feed) to offset fecal sludge collection and processing costs.

Finally, pro-poor sanitation programs continue to include the development of sewerage options. The Kenya Informal Settlements Improvement Project (KISIP) financed the expansion of sewer connections in two low-income areas in the city of Kisumu [15]. Also in Kenya, the Nakuru Water and Sanitation Services Company, the water and sewerage utility for the city of Nakuru, partnered with WSUP to construct a sewer network in one of the city's low-income areas [4]. In Brazil, condominial sewerage systems connect housing blocks to small dispersed treatment facilities, thereby reducing the need to build and maintain larger-scale traditional infrastructure [16].

To support sanitation improvements in Kisumu, the third largest city of Kenya, we measured the gap between market prices and WTP for safe emptying services in the city's low-income areas. Stated and revealed willingness-to-pay (WTP) methods provide options for quantifying demand for improved sanitation products and services among low-income households. Stated WTP methods generally use household surveys to assess demand for hypothetical products and services by asking respondents to reflect on price points through a series of questions [1, 17–19]. In contrast, revealed WTP methods obtain results from real money price-responses (i.e., market data or experiments) [17] and are generally considered more reliable because they reflect respondents' actual (rather than hypothetical) purchasing behaviors [17]. Our study, therefore, addressed the following objectives: 1) to compare household demand for safe emptying services with the prices charged for these services; and 2) to compare stated and revealed WTP by conducting both household surveys and a real-money voucher experiment.

## Methods

### Study site

Kisumu is Kenya's third largest city with an estimated 419,000 inhabitants in its urban constituencies, of which approximately 60% live in low-income areas [20, 21]. Approximately 20% of households are connected to the sewerage network that transports waste to two treatment facilities operated by the local utility, Kisumu Water and Sewerage Company (KIWASCO) [21]. An estimated 70% of households use pit latrines, and 5% use septic tanks, primarily in middle- and high-income households [21]. Only 33% of Kisumu's fecal waste (sewage and fecal sludge) is safely managed, mostly via treatment at KIWASCO's fecal sludge treatment site [21]. In Kisumu's low-income areas, it is common for informal manual pit emptiers to dispose of fecal waste in nearby waterways or bury it onsite [22].

### Study design

Our study design comprised two phases. From May to November 2018, we conducted a market assessment to identify existing emptying services in Kisumu and to document barriers that prevented safe pit-emptying services from operating in low-income areas. From February to June 2019, we conducted a household survey and real-money voucher trial to evaluate the WTP of Kisumu's low-income residents for safe pit-emptying services by licensed operators that removed and transported fecal sludge for offsite treatment.

### Market assessment

Our market assessment included the following activities: i) literature review of national and county-level policy and program documents relevant to sanitation; ii) in-depth interviews with key water and sanitation stakeholders; iii) pit emptying observations; iv) transect walks in low-

income areas; and (v) focus group discussions with low-income residents (Fig 1). Our literature review examined national strategies and policies, the county's draft sanitation policy, and documentation on county and city sanitation programs. We interviewed multiple stakeholders in Kisumu's sanitation sector using a 'snowball' method [23], whereby we started with contacts at Water and Sanitation for the Urban Poor (WSUP), an international non-governmental (NGO) organization implementing programs in Kisumu, and KIWASCO who subsequently referred us to other stakeholders. These informants included additional KIWASCO staff, municipal and county government officials, microfinance institution representatives, NGO workers, and fecal sludge emptying service providers. The fecal sludge emptying service providers included vacuum truck operators (VTOs), informal manual emptiers, and "formal" manual emptiers; we defined formal manual emptying as trained emptying groups that worked with recognition of the local government and National Environment Management Authority (NEMA), had permission to dump waste at the city's treatment facility, and complied with public health regulations. We observed pit emptying by these service providers to understand their current operating processes. We also conducted transect walks to document contextual characteristics of the neighborhoods where they operated, such as access to water and sanitation services, geological and infrastructure conditions, typology (urban, peri-urban, rural) and land tenure (private versus government housing). Additionally, we held Focus Group Discussions (FGDs) in several low-income areas to understand sanitation and emptying preferences and challenges. We recruited FGD participants through local leaders (e.g., local village elders or chiefs) identified through WSUP and held separate FGDs for persons with disabilities and women. We conducted interviews and FGDs in either the local languages (Swahili and Luo) or English; we took detailed notes in English. We conducted interviews, emptying observations, and FGDs until we reached saturation (i.e., no new information was obtained with additional data collection).

## Pit-emptying stakeholders and service providers

The Kisumu County Government and the National Environment Management Authority (NEMA) are the main official bodies responsible for sanitation policy implementation and regulation in the city of Kisumu. At the county level, NEMA is responsible for monitoring environmental impact assessments and effluent discharged into sewerage systems [24]. The Kisumu County Government ensures that public health standards are met and is responsible for licensing VTOs. In practice, however, NEMA and the city Public Health department facilitate VTO licensing. KIWASCO is the sole utility providing water and sewerage services in the city of Kisumu. NGOs also play a role in the provision of sanitation facilities and services. WSUP, for example, developed standard operating procedures for safe fecal sludge management and supported Kisumu County in the development of the Environmental Sanitation and Hygiene Policy. WSUP also trained Gasia Poa, one of the three formal manual pit-emptying groups that primarily operate in low-income areas of Kisumu city, on safe emptying practices in 2018 (Table 1). Gasia Poa has established a licensing agreement with the County Public Health Office and is allowed to dispose of fecal waste at KIWASCO wastewater treatment facilities free of charge. Another program, KisumuSan (2016–2021), led by Practical Action, trained the two other manual emptying groups (Vukasasa in the Nyalenda low-income area and BlueStars in the Obunga low-income area) and provided each group with a portable mechanized emptying device called a Gulper (Table 1). During our study period, eight vacuum trucks were operating in Kisumu City: KIWASCO owned two vacuum trucks and the remaining six were privately owned. Only four of the vacuum trucks (two from KIWASCO and two that were private) were licensed by NEMA. Finally, informal, unlicensed, manual emptiers

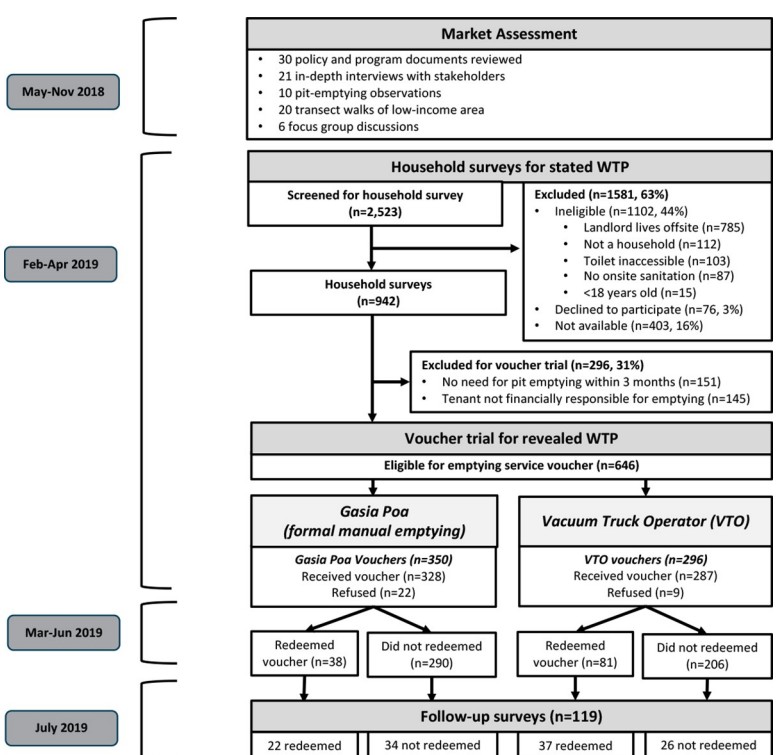

**Fig 1. Study design.** Household survey participant flow and voucher distribution. (WTP = willingness-to-pay; VTO = Vacuum Truck Operator).

also commonly operate in low-income areas of Kisumu city, though typically at night to avoid penalties [4]. Table 1 describes the different emptying groups with their respective price structures.

## Study population

Table 2 summarizes the characteristics of survey respondents and those eligible for vouchers. We conducted surveys with 942 households and allocated vouchers to 646 (69%) of these households (Fig 1). Characteristics were generally similar between survey respondents and the subset of the population that were eligible for vouchers (Table 2). Approximately half (53%, 500/939) of the respondents had at least some secondary education. The majority of respondents (75%, 704/942) were landlords or homeowners, due to our sampling strategy targeting this population, which is described in the Methods section 2.4.3. Households had a median of five members and 42% (392/942) had a least one child under five years of age. When asked about their monthly household income ranges, 29% (258/892) of households reported <7,000 KES (70 USD), and 23% (206/892) of households reported 7,000 to <10,000 KES (70 to <100 USD), indicating that at least half of the population was living on less than 100 KES (1 USD) per person per day. The median time living on the compound was 18 years.

Most households (74%, 698/942) had dry improved pit latrines; the remainder (19%,176/942) primarily had pour-flush toilets connected to some form of underground containment (Table 2). Most toilet facilities (85%, 801/942) were shared, serving a median of six households. Other reports note that across the entire city, approximately 30% of households share toilets [21]. Landlords/homeowners almost always paid for their own latrine construction (99%, 699/704) and tenants rarely paid for latrine construction (5%, 34/733). Piped water supplied by

**Table 1. Key emptying service providers in Kisumu.**

| Emptying group | Description | Market prices [a] |
|---|---|---|
| VTOs | • Eight vacuum trucks operate in Kisumu (privately-owned or managed by KIWASCO).<br>• VTOs primarily serve middle- and high-income households.<br>• VTOs must be able to access households via roads.<br>• VTOs are unable to handle solid waste often found in underground containment. | Pit latrine: **6,000 KES** [b, c]<br><br>Septic tank: **4,000 KES** [b, c] |
| Formal manual emptiers | • Three groups operate in Kisumu: Gasia Poa, Vukasasa, and BlueStars.<br>• NGOs (WSUP, Umande Trust, Kisumu Urban Apostolate Programmes (KUAP) and Practical Action) trained all three groups on safe practices and provided some emptying equipment.<br>• All three groups safely dispose of fecal waste at KIWASCO fecal sludge treatment site. One of them, Gasia Poa, can do so free of charge.<br>• Formal emptiers have somewhat limited activity; prior to our research, emptying groups emptied approximately 5–10 toilets per month.<br>• These groups charge an additional flat fee for solid waste removal from pit latrines. | Pit latrine: **9,000–12,000 KES** [b, d]<br><br>Septic tank: **7,000 KES** [b, d] |
| Informal manual emptiers | • This is the primary form of pit emptying in low-income areas.<br>• Informal manual emptiers dump fecal waste into drains or bury it onsite.<br>• They work without proper equipment, training or oversight, and typically at night.<br>• Weak regulation and enforcement enable informal emptiers to operate without licenses. | Pit latrine: **1,000–3,000 KES** |

[a] Exchange rate of 100 KES = 1 USD (oanda.com, June 6, 2019)

[b] Prices may vary based on distance to treatment facility. Prices are generally for one trip to the fecal sludge treatment facility for safe disposal, and larger underground containment may require multiple trips.

[c] A typical vacuum exhauster truck can hold approximately 8,000 liters of fecal sludge.

[d] Prices listed are for the formal manual emptying group Gasia Poa; prices for the other two groups were comparable (6,000–7,500 KES per pit latrine toilet door, and 7,000 KES for septic tanks for Vukasasa; 8,000–12,000 KES for BlueStars).

KIWASCO was the primary drinking water source for most households: either via household taps (44%), neighbor or shared connections (30%), or public taps (20%).

We found that 40% (374/942) of households had emptied their underground containment at least once within the past 12 months, and 16% (147/942) had emptied in the past three months (Table 2). FGD respondents noted that some households emptied their underground containment as often as every one to two months, especially during the rainy season and in areas with high water tables, where pits are usually shallow (approximately 1.2–2.4 m) and fill up quickly; flooding, which is common in low-income areas in the east of Kisumu (such as Nyalenda and Manyatta), also increased pit-emptying requirements.

Household survey respondents reported paying a median of 3,000 KES (30 USD) for a single emptying, including solid waste removal (Table 2). Most households (62%) employed informal manual emptiers, though 28% of households contracted with an operator of a small vacuum truck ($<7 m^3$) (Table 2). In both household surveys and FGDs, respondents reported that they did not use formal manual emptiers because they were unfamiliar with the service. Respondents reported not using VTOs because they perceived that VTO services were too expensive, could not access their latrines, and/or would not empty pits that contained solid waste. Approximately half of tenants (53%, 67/126) paid for their own containment emptying (Table 2).

**Table 2. Household characteristics for survey respondents (n = 942) and household eligible for vouchers (n = 646).**

| Category | Survey Respondents | | Voucher Recipients | |
| --- | --- | --- | --- | --- |
| | N = 942 | | N = 646 | |
| | Percentage | N | Percentage | N |
| Gender (% Female) | 60% | 566 | 56% | 361 |
| Median age (interquartile range)[a] | 40 (31–53) | | 41 (32–56) | |
| Education[a] | | | | |
| None | 6% | 60 | 7% | 47 |
| Primary | 40% | 379 | 40% | 260 |
| Secondary | 33% | 308 | 32% | 208 |
| College/University | 20% | 192 | 20% | 128 |
| Married | 66% | 618 | 66% | 426 |
| Median household size (interquartile range) | 5 (3–6) | | 5 (3–7) | |
| Households with children <5 years | 42% | 392 | 40% | 259 |
| Home ownership (includes landlords) | 75% | 704 | 89% | 575 |
| Median years living in compound (interquartile range)[a] | 18 (6–30) | | 20 (10–33) | |
| Monthly household income (KES)[a,b] | | | | |
| <5,000 | 15% | 138 | 15% | 90 |
| 5,000-<7,000 | 13% | 120 | 13% | 81 |
| 7,000-<10,000 | 23% | 206 | 23% | 139 |
| 10,000-<23,000 | 32% | 283 | 33% | 202 |
| ≥23,000 | 16% | 142 | 16% | 98 |
| Have M-Pesa (i.e., mobile money payment service) | 97% | 909 | 96% | 621 |
| Sanitation type[c] | | | | |
| Dry improved pit latrine | 74% | 698 | 75% | 484 |
| Pour-flush to septic tank | 12% | 110 | 10% | 65 |
| Dry unimproved pit latrine | 7% | 68 | 8% | 50 |
| Pour-flush to pit latrine | 7% | 66 | 7% | 47 |
| Sharing toilet facility | 85% | 801 | 87% | 562 |
| Water source | | | | |
| Piped (private) | 44% | 414 | 45% | 290 |
| Piped (neighbors/shared) | 30% | 283 | 32% | 206 |
| Public tap | 20% | 185 | 17% | 109 |
| Well (protected or unprotected) | 5% | 49 | 6% | 36 |
| Other[d] | 1% | 11 | 1% | 5 |
| When last emptied | | | | |
| Within 3 months ago | 16% | 147 | 15% | 97 |
| Between 3 and 12 months ago | 24% | 227 | 27% | 176 |
| 1–2 years ago | 11% | 107 | 15% | 94 |
| Over 2 years ago | 6% | 55 | 7% | 44 |
| Never | 40% | 376 | 35% | 228 |
| Unknown | 3% | 30 | 1% | 7 |
| Who last emptied the containment[e] | | | | |
| Illegal manual emptier | 62% | 332 | 65% | 268 |
| Small vacuum truck (<7m[c]) | 28% | 149 | 25% | 101 |
| Large vacuum truck (7-10m[c]) | 4% | 23 | 4% | 15 |
| Formal manual emptier | 4% | 22 | 5% | 20 |
| Other[e] | 2% | 10 | 2% | 7 |
| Paid for their own last emptying[f] | | | | |

(*Continued*)

**Table 2.** (Continued)

| Category | Survey Respondents | | Voucher Recipients | |
|---|---|---|---|---|
| | N = 942 | | N = 646 | |
| | Percentage | N | Percentage | N |
| Homeowners/landlords | 99% | 377 | 99% | 342 |
| Tenants | 53% | 67 | 100% | 42 |
| Median cost of emptying (KES) (interquartile range) | 3000 (2500–4000) | | 3000 (2500–4000) | |

[a] Data are missing for all survey respondents for age (2), education (3), years living in compound (3), household income (50), time to water source (1), and cost of water (3). Data are missing for voucher recipients for education (3), years living in compound (3), household income (36), time to water source (1), and cost of water (2). Income data is missing because households didn't know or refused to answer.

[b] Exchange rate of 100 KES = 1 USD (oanda.com, June 6, 2019).

[c] Improved pit latrines were defined as pit latrines with a concrete or plastic slab, ventilated improved pit latrines, or pit latrines with a sitting toilet. Unimproved pit latrines were defined as pit latrines with mud or wood flooring, or an open pit. We confirmed sanitation type (dry vs. pour flush) and flooring through observation, but we were unable to confirm the type of underground containment (septic tank vs. pit), and therefore relied on reported responses.

[d] For all survey respondents, other water sources included water vendor (8), rainwater (2), unprotected spring (1). For voucher recipients, other water sources included water vendor (4) and rainwater (1).

[e] Data for households that knew when their pit latrine was last emptied: n = 536 survey respondents and n = 411 for voucher recipients. Other includes emptied by self/family member (3 survey respondents, 2 voucher recipients), canter truck with pump (2 survey respondents, 1 voucher recipients), or unknown (5 survey respondents, 4 voucher recipients).

[f] Data for households that knew when their pit was last emptied and who paid; n = 506 survey respondents (380 homeowners/landlords and 126 tenants) and n = 386 for voucher recipients (344 homeowners/landlords and 42 tenants).

## Demand assessments

**Study sites.** We conducted the household surveys and voucher trial in eight low-income areas of Kisumu (Bandani, Manyatta A, Manyatta B, Nyabera, Nyalenda A, Nyalenda B, Nyamasaria, and Obunga) [25]. These eight areas met the following criteria for the household surveys: i) located within urban Kisumu (including Central, East, and West constituencies); ii) classified as urban (i.e., not peri-urban or rural) as observed during transect walks; iii) did not have government housing according to local sources (i.e., KIWASCO, WSUP); and iv) had a limited number of sewer connections (estimated <20% coverage).

**Emptying services.** We measured WTP for two existing safe pit-emptying services: i) Gasia Poa, a formal manual emptying group trained by WSUP that deployed emptiers equipped with safety gear to manually transfer waste from pit latrines and septic tanks to large barrels; and ii) VTOs that used exhauster trucks to suction fecal waste from pit latrines and septic tanks. Both services transported fecal sludge to one of KIWASCO's treatment sites and were licensed by the local authorities. Gasia Poa also provided manual emptiers with vaccinations against infectious diseases and protective gear/equipment to avoid direct contact with fecal sludge. We explained these emptying services to study participants with detailed graphics (S1 Fig). We did not measure WTP for informal manual emptying because the price that study households paid for informal emptying was well established at 1,000–3,000 KES (Table 2).

**Sampling.** We randomly selected households to participate in our surveys, which collected data on household demographics, socioeconomic status, water and sanitation access, and WTP for emptying services. We used ArcGIS (ESRI, Redlands, CA, USA) to randomly select a number of GPS coordinates proportional to each low-income area's total population [26]. Survey enumerators navigated to each GPS coordinate and identified the nearest compound. Subsequently, they interviewed one landlord or homeowner in the compound, if available, and a single tenant (if multiple tenants were present, the enumerator selected the first one

on the left side of the compound). Enumerators continued surveying all compounds in one designated direction until they reached a maximum of five surveys per day (four landlords or homeowners and one tenant). We designed this sampling strategy so that approximately 80% of respondents were landlords or homeowners and 20% were tenants, because the financial responsibility for emptying generally falls on landlords and homeowners (i.e., tenants typically depend on landlords to pay for emptying) [22]. Households were eligible to participate in the household survey if they: i) were exclusively a residence (not a business or institution); ii) had an adult (≥18 years) head of household; iii) had an onsite sanitation facility that required emptying; and iv) were accessible to emptying by service providers. Additionally, only landlords that lived on the premises were eligible to participate. If a head of household was not available during the initial visit, the enumerators returned to the household at least two more times before classifying the household as not available. The questionnaire was administered in the local language, Luo (S1 and S2 Texts).

**Stated willingness-to-pay.** WTP estimates obtained through surveys are referred to as *stated* preferences [17]. Our household survey employed the double-bounded dichotomous choice method to measure stated WTP for pit-emptying services [27]. We first asked respondents to provide a "yes/no" answer to two closed-ended price probes [18, 28] (Fig 2). If the respondent answered "yes" to the first question, we then queried their willingness to pay a higher price. Alternatively, if the respondent answered "no" to the first question, we then queried their willingness to pay a lower amount. Providing a sequence of questions motivates respondents think about a relevant price range [1, 18, 19], and other studies of WTP for water and sanitation products and services have applied similar dichotomous choice methods [6, 8, 19, 28, 29]. We followed the dichotomous choice questions with an open-ended question asking respondents to state the highest amount they would be willing to pay [30]. We report this maximum WTP as "stated WTP" in the rest of the paper.

In each survey, we randomized the starting price points for the double-bound dichotomous choice questions (Fig 2). We established these price points at approximately 20%, 40%, 60%

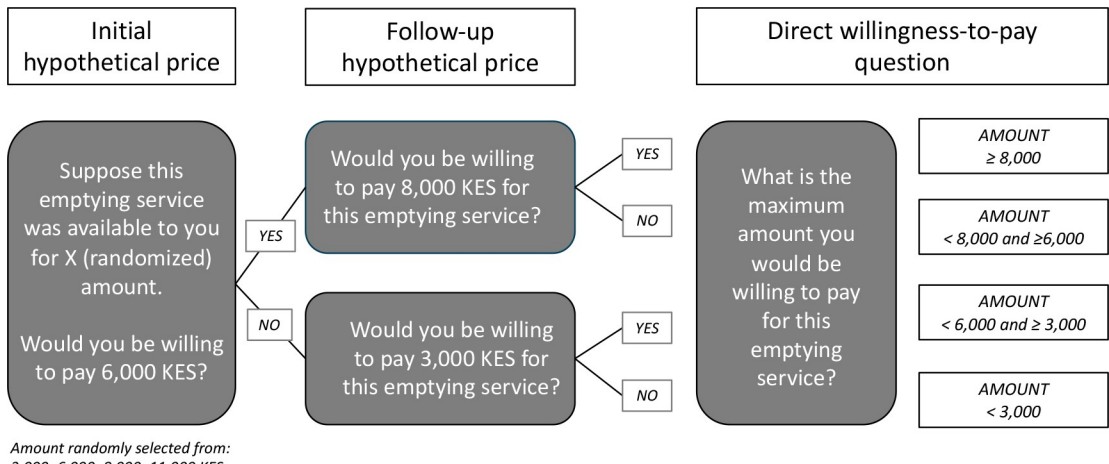

*Amount randomly selected from: 3,000; 6,000; 8,000; 11,000 KES*

**Fig 2. Sequence of willingness-to-pay questions.** Using the double-bounded dichotomous choice method, we randomly asked households (i.e. landlords/homeowners and tenants) if they were willing to pay 3,000/6,000/8,000/11,000 KES (30/60/80/110 USD) for Gasia Poa emptying services. Depending on whether the respondent answered positively or negatively to the randomized amount, we then asked participants the same question for the next higher or lower amount. For example, if a respondent was willing-to-pay 6,000 KES (60 USD), we then asked if she was willing-to-pay 8,000 KES (80 USD), followed by an open-ended question about their maximum WTP. We used the same method to assess WTP for VTO emptying services with different starting price points.

and 80% of market prices of emptying services (rounding to the nearest 500 or 1,000 KES); we estimated market prices to be 14,000 KES (140 USD) for Gasia Poa (inclusive of an additional fee for solid waste removal), 6,000 KES (60 USD) for VTO emptying of dry pits, and 4,000 KES (40 USD) for VTO emptying of wet pits or septic tanks. As a result, our randomized starting price points were: 3,000/6,000/8,000/11,000 KES (30/60/80/110 USD) for Gasia Poa emptying services; 2,000/3,000/4,000/5,000 KES (20/30/40/50 USD) for VTO emptying of dry pits; and 500/1,500/2,500/3,500 KES (5/15/25/35 USD) for VTO emptying of wet pits. We used the VTO price points for wet pits if households had flush/pour flush latrines; otherwise, we used the dry pit price points. We queried all respondents regarding WTP for both Gasia Poa and VTO emptying services.

We also asked all respondents about their WTP for subscription emptying services for Gasia Poa and VTOs, as an alternative to large one-time household outlays. We presented these subscription services as twelve monthly payments that would cover one emptying service during the twelve-month period, at the time of the subscriber's choice. Prior studies have demonstrated that tenants' demand for water and sanitation services is reflected in rent premia [22, 31, 32]. Therefore, when surveying tenants, we asked about their WTP for an increase in rent (rather than WTP for one-time outlay or subscription services) if emptying services were provided when necessary in their dwelling. Our starting price points for rent increases were 50/100/200/300 KES (0.5/1/2/3 USD) per month, for both Gasia Poa and VTO services.

**Revealed willingness-to-pay.** Results obtained from real money price-responses (i.e., market data or experiments) are often referred to as *revealed* preferences [17]. Following the household survey, we conducted a real-money sales trial to estimate revealed WTP for the two types of emptying services. To conduct our sales trial, we randomly distributed vouchers that provided discounted prices for pit-emptying services to eligible households. Subsequently, we tracked voucher redemption rates at each discount level to estimate WTP at different price points (Fig 1). We selected households for the real-money sales trial if they responded to the household survey, they reported that they were financially responsible for pit emptying, and their pit would likely require emptying at any time within three months of the survey. We classified households as likely to require pit emptying within the next three months if they met any of the following criteria: (i) they reported that they intended to empty their pit in the next three months, (ii) enumerators observed that fecal sludge was within one meter from the top of the pit, (iii) there were at least 25 households living in the same compound, (iv) their underground containment had not been emptied within the last two years, or (v) they were living in a high-water-table area and had lived there less than one year (and therefore were presumably unaware of the frequency of emptying needed); 97% of our study population met the first or second criteria. We selected the three-month time frame based on our market assessment results: households reported emptying underground containment every one or two months in areas with high water tables and every six months in areas with low water tables.

We provided eligible households with a discount voucher for either VTO or Gasia Poa emptying services based on the following factors: households were eligible for Gasia Poa vouchers if their superstructure and slab were stable enough to support Gasia Poa emptying methods, and households were eligible for VTO services if they were located within 50 meters of a road. Households eligible for both Gasia Poa and VTO services randomly received a voucher for one of the two services. As previously noted, we also classified VTO services as wet (vs. dry) if they had flush/pour-flush latrines; the VTOs verified this classification and pit latrine accessibility before vouchers were distributed. If multiple latrines were eligible for emptying within a single household compound, enumerators asked the respondent to select one latrine for emptying with the discount voucher.

We established the voucher price points at approximately 20%, 40%, 60% and 80% of market prices of emptying services (rounding to the nearest 500 or 1,000 KES); we estimated market prices to be 14,000 KES (140 USD) for Gasia Poa (inclusive of an additional fee for solid waste removal), 6,000 KES (60 USD) for VTO emptying of dry pits, and 4,000 KES (40 USD) for VTO emptying of wet pits or septic tanks. As a result, the discounted voucher price points were: 3,000/6,000/8,000/11,000 KES (30/60/80/110 USD) for Gasia Poa emptying services; 2,000/3,000/4,000/5,000 KES (20/30/40/50 USD) for VTO emptying of dry pits; and 500/1,500/2,500/3,500 KES (5/15/25/35 USD) for VTO emptying of wet pits. The vouchers included a household identification number, a phone number for the pit-emptying service, the estimated market price for the specified service, the discount amount that the household received, and their expiration date (S2 Fig). We instructed households that they should not exchange vouchers with other households, and we marked eligible latrines with stickers specifying the service provider's name and contact details. Households had approximately two to four months to redeem vouchers before their expiration on June 6, 2019.

To redeem their discount vouchers, households had to call a phone number on the voucher to coordinate the pit-emptying service. Households paid for their discounted emptying services via the mobile money application M-Pesa. To monitor services provided in return for voucher redemptions, we attended 71% (85/119) of the emptying jobs. We reimbursed pit emptier groups the difference between the market price of the emptying service and the discount voucher amount.

In July 2019, approximately 3–5 months after discount voucher distribution, we conducted follow-up surveys with approximately 20% of the households that received discount vouchers. We stratified this sample by voucher redemption so that 50% of follow-up survey respondents redeemed their vouchers (Fig 1). Within each stratum, we randomly selected a subset of households for follow-up using Excel.

**Sample size.** Our target sample size was 300 vouchers per emptying service, or 75 vouchers per price point (600 vouchers total). This sample size allowed us to detect a minimum 22 percentage-point difference in the proportion of the population willing to pay for the two different services at each price point.

## Data analysis

During our market assessment, we analyzed our interview notes for common themes and triangulated information from key informant interviews, FGDs, and pit emptying observations. We entered household survey responses into the CommCare survey and data management application (DiMagi Inc., Cambridge, MA USA) on mobile phones (Samsung Galaxy J4, South Korea). We conducted quality control checks on 10% (95/942) of surveys: 4.6% (43/942) spot checks and 5.5% (52/942) back checks. When calculating revealed WTP, we classified households that were eligible for vouchers but refused their randomly allocated discount voucher as not willing to pay the voucher price (5%, 31/646 households) (Fig 1). We categorized participating households into socioeconomic status quintiles using an asset index and principle component analysis [33]. To generate customer demand curves, we used the cumulative percentage of respondents willing to pay for emptying services at different prices. To compare WTP for monthly subscription vs. one-time payments, we adjusted the 12 monthly payments to their net present value using a discount rate of 10%. To examine factors associated with WTP, we conducted bivariate analyses of mean stated WTP using sample t-tests for binary variables and an analysis of variance (ANOVA) for categorical variables (e.g., wealth quintiles and education levels). We also examined factors associated with WTP in multivariate models using maximum likelihood functions for the double-bound dichotomous choice method [8]. To

examine factors associated with voucher redemption, we conducted bivariate and multivariate analyses using logistic regression. For all types of multivariate models, we included all factors with bivariate associations with a significance value of $p < 0.1$. We tested for collinearity by computing the condition index from the correlation matrix and excluded variables if their condition index was >10 [34]. Furthermore, we compared WTP data for stated and revealed preferences using a linear regression model controlling for the survey or voucher price point. We analyzed quantitative data using the statistical package Stata 15 (StataCorp, College Station, TX). We used the following exchange rate for the analysis: 100 KES to USD 1.00 (June 6, 2019, oanda.com).

## Financial requirements for expanding safe emptying services

Approximately 62% of Kisumu's population, or 51,965 households, currently lack safe emptying & transport services [21]. Even with the projected sewer expansion to an additional 20% of households [35], 42% of the population, or an estimated 35,202 households, will continue to require safe emptying & transport services in the near- to mid-term future (S1 Table). Using data from our market assessment and household surveys, we estimated the financial requirements for serving these households, both in terms of capital requirements and operational costs under two scenarios: with and without the projected sewer expansion. All assumptions and step-by-step calculations are provided in S1 Table. In particular, we assumed that 75% of toilets would be serviced by VTOs and 25% by formal manual emptiers, based on a complementary study in which we assessed the expansion potential of both services in Kisumu's low-income neighborhoods [36]. To calculate capital expenditures (CAPEX), we estimated the number of exhauster trucks, pick-up trucks and tractors required to serve an additional 35,202 or 51,965 households, depending on the scenario (S1 Table). We approximated operational expenditures (OPEX) using average market prices, which capture capital depreciation and possible service provider margins. To estimate household subsidy requirements, we compared OPEX to the amount that approximately 80% of households were willing to pay for safe emptying services (S1 Table). We verified that this amount was consistent with current market prices for informal manual emptying (Table 1).

## Research approvals and ethical review

The Western Institutional Review Board (WIRB) (Puyallup, WA, USA) determined this study exempt from full ethical review under 45 CFR §46.101(b)(2) of the Federal Common Rule in the USA. In Kenya, we obtained ethical approval for our research from Amref Health Africa (AMREF) (ESRC P493/2018) and a research permit from the National Commission for Science, Technology, and Innovation (NACOSTI) (NACOSTI/P/19/39980/28701). We provided all participants with verbal and printed details of the study in the local language. We also obtained informed written and verbal consent from all participants.

## Results

### Stated WTP from household survey

Stated WTP was higher for Gasia Poa services than for VTO services (Fig 3A). Median stated WTP was 3,000 KES (30 USD) for Gasia Poa services (mean = 3,438 KES; 34 USD), representing approximately 25–43% of market prices (which ranged from 7,000–12,000 KES, 70–120 USD). Less than 20% of respondents were willing to pay the lower-bound market price of 7,000 KES (70 USD) (Fig 3A). Median stated WTP was 2,000 KES (20 USD) for VTO services (mean = 2,182 KES, 22 USD), representing approximately 33–50% of market prices (which

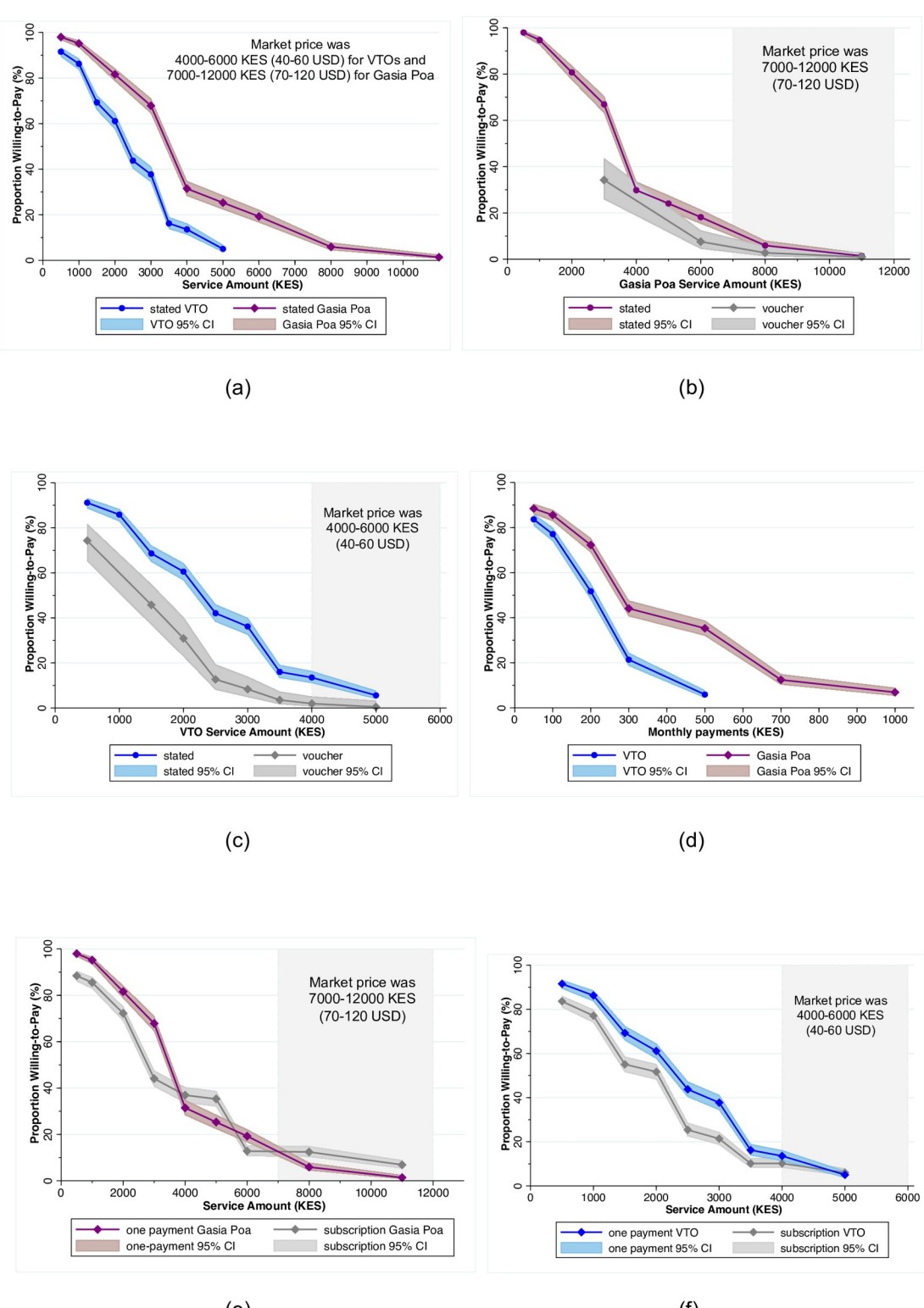

**Fig 3. Willingness-to-pay.** (a) Stated WTP for Gasia Poa and VTOs (n = 942), (b) Stated and revealed (voucher) WTP for Gasia Poa (n = 646), (c) Stated and revealed (voucher) WTP for VTOs (n = 646), (d) Stated WTP for monthly subscription services for Gasia Poa (n = 795) and VTOs (n = 796) (e) Stated WTP for Gasia Poa comparing one-time and subscription services (n = 795), (f) Stated WTP for VTOs comparing one-time and subscription services (n = 796), and (g) Stated WTP for additions to rent (n = 145). Exchange rate of 100 KES = 1 USD (oanda.com, June 6, 2019).

ranged from 4,000–6,000 KES, 40–60 USD). 14% of respondents had a stated WTP of 4,000 KES (40 USD), the lowest market price for VTO services (Fig 3A).

### Revealed WTP from voucher trial

Households redeemed 119 vouchers, which represented 11% (38/350) of the vouchers distributed for Gasia Poa emptying services and 27% (81/296) of the vouchers distributed for VTO emptying services (Fig 1). Revealed WTP was lower than stated WTP for both Gasia Poa and VTO services at every price point; however, for Gasia Poa services, this difference was not statistically significant across the comparable price range ($p = 0.17$) (Fig 3B). We did not provide Gasia Poa vouchers at prices under 3,000 KES (30 USD) and, therefore, we were unable to compare revealed and stated demand at prices lower than this amount (Fig 3B). With respect to VTO services, the proportion of respondents that redeemed discount vouchers was 39 percentage points less than the proportions that stated they would pay those amounts during our surveys ($p < 0.01$) (Fig 3C).

### WTP for subscription services and rent increases

Based on the household survey data, we determined that demand for a VTO service subscription ranged from 52% of respondents willing to pay 200 KES (2 USD) per month to 6% of respondents willing to pay 500 KES (5 USD) per month (Fig 3D). Stated WTP for a subscription fee was again higher for emptying services provided by Gasia Poa, ranging from 44% of households willing to pay 300 KES (3 USD) per month to 12% of households willing to pay 700 KES (7 USD) per month (Fig 3D). Across price points, we did not find substantially higher stated WTP among households when comparing the annual (12-month) sum of monthly subscription payments (using net present value) with one-time services payments for either emptying by VTOs or Gasia Poa (Fig 3E and 3F). However, when we asked respondents if they preferred a 12-month subscription service compared to a one-time payment, 61% (487/942) reported that they preferred the subscription (38% preferred the one-time payment and 1% preferred a combination of the two).

Tenant willingness to increase rent payments to cover safe emptying services was almost identical for VTO and Gasia Poa services (Fig 3G). The median maximum increase that respondents stated being willing to pay was 100 KES (1 USD) per month for both services. At an increase of 50 KES (0.5 USD) per month, 90% of tenants were willing to pay for VTO or Gasia Poa services. At 100 KES (1 USD) per month, 78% were willing to pay for Gasia Poa and 80% for VTOs; at 300 KES (3 USD) per month, the proportion dropped to 23% for Gasia Poa and 20% for VTOs (Fig 3G). For comparison, tenants reported paying a median rent of 3,000 KES (30 USD) per month.

### Factors associated with WTP

Bivariate and multivariate model results showed that male and younger survey respondents were willing to pay significantly more for both types of emptying services (S2 Table and S3 Table, all $p < 0.05$). In the multivariate models, survey respondents had a significantly higher stated WTP for Gasia Poa services if they reported sharing a latrine ($p = 0.03$) and were more educated ($p = 0.03$) (S3 Table). Survey respondents had a significantly higher stated WTP for VTO services if they were wealthier ($p < 0.01$), were tenants ($p < 0.01$), and had a toilet located less than 50 m from a road ($p < 0.01$) (S2 Table and S3 Table). With respect to voucher redemption, we found that respondents were significantly more likely to redeem their vouchers if they were wealthier ($p < 0.01$) and received a higher discount voucher ($p < 0.01$) (S4 Table).

## Follow-up surveys

To obtain customer feedback, we conducted follow-up surveys with 119 households that received vouchers: 59 households that redeemed their vouchers and 60 that did not redeem. We found that most households that redeemed their vouchers were satisfied with the emptying service: when asked if they would use the same service again, 84% of households responded positively (Table 3). When asked about future emptyings, 54% of households reported that they planned to use VTOs, and 46% planned to use a formal manual emptying group; most households chose the service they had used through voucher redemption and none of the households reported plans to use informal manual emptiers (Table 3). When asked what they liked about the emptying services, common responses were that the service had minimal smell (51%), no spillage (34%), solid waste removal (32%), proper sludge disposal (32%), and/or the entire pit was emptied (29%) (Table 3). When asked what they disliked about the service,

**Table 3. Follow-up survey with voucher redeemers.**

| | Gasia Poa voucher (n = 22) | | VTO voucher (n = 37) | | Total (n = 59) | |
|---|---|---|---|---|---|---|
| | Percentage | N | Percentage | N | Percentage | N |
| Would use same service again[a] | | | | | | |
| Yes, this same service | 90% | 19 | 80% | 28 | 84% | 47 |
| Yes, but a different operator | 10% | 2 | 9% | 3 | 9% | 5 |
| No | 0% | 0 | 11% | 4 | 7% | 4 |
| Preferred service for next emptying[a] | | | | | | |
| VTOs | 0% | 0 | 86% | 30 | 54% | 30 |
| Gasia Poa | 100% | 21 | 14% | 5 | 46% | 26 |
| How paid for service[a] | | | | | | |
| Savings | 77% | 17 | 89% | 31 | 81% | 48 |
| Borrowed | 14% | 3 | 3% | 1 | 7% | 4 |
| Credit | 9% | 2 | 6% | 2 | 7% | 4 |
| Other[b] | 9% | 2 | 3% | 1 | 5% | 3 |
| Liked about the service | | | | | | |
| Minimal odor | 73% | 16 | 38% | 14 | 51% | 30 |
| No spillages | 45% | 10 | 27% | 10 | 34% | 20 |
| Solid waste removed | 64% | 14 | 14% | 5 | 32% | 19 |
| Proper sludge disposal | 50% | 11 | 22% | 8 | 32% | 19 |
| Entire pit emptied | 36% | 8 | 24% | 9 | 29% | 17 |
| Reasonable price | 5% | 1 | 35% | 13 | 24% | 14 |
| Quick service | 0% | 0 | 22% | 8 | 14% | 8 |
| Came quickly after scheduling | 14% | 3 | 11% | 4 | 12% | 7 |
| Other[c] | 27% | 6 | 19% | 7 | 22% | 13 |
| Disliked about the service | | | | | | |
| Pit not fully emptied | 14% | 3 | 22% | 8 | 19% | 11 |
| Too costly | 5% | 1 | 16% | 6 | 12% | 7 |
| Solid waste not removed | 0% | 0 | 16% | 6 | 10% | 6 |
| Other[d] | 5% | 1 | 30% | 11 | 20% | 12 |

[a] Data missing for using service again (1 Gasia Poa, 2 VTOs), preferred service (1 Gasia Poa, 2 VTOs), payment data (2 VTOs).

[b] Other includes borrowed from mobile platform (1), paid for by relative (1), and part of the contribution came from other houses.

[c] Other reasons included quiet service (2 Gasia Poa, 1 VTOs), pleasant employees (1 Gasia Poa, 3 VTOs), came promptly (3 Gasia Poa, 3 VTOs), no flies (1 Gasia Poa).

[d] Other reasons for disliking the service included spills (1 VTO), odor (3 VTOs), noisy (3 VTOs), service took too long (3 VTOs), came late (1 VTO), price disagreement (1 Gasia Poa).

common responses were that their pits were not fully emptied (19%), the service was too costly (12%), or solid waste was not removed (10%) (Table 3).

Among households that did not redeem their emptying vouchers, the main barriers for voucher redemption that they reported were lack of money (25/60, 42%) or high price (15/60, 27%). Ten households (17%) reported that their pits did not require emptying, though five of these households also listed lack of money or high price as barriers. Other reasons (all reported by fewer than five households) were that households were not sure how to redeem their vouchers, had difficulty scheduling the service, were waiting for the service provider to follow up, had plans to upgrade their latrine, lost their voucher, or lived near road construction that made the household inaccessible. Fifteen of the households that did not redeem their vouchers (25%) reported using another service to empty their pit in the past three months: nine reported using informal manual emptiers, four reported using VTOs, and two reported using one of the formal manual emptying groups.

### Financial requirements for expanding safe emptying services

To expand formalized fecal sludge emptying and transportation services to the approximately 51,965 households of Kisumu that are currently underserved (corresponding to approximately 8,661 toilets), we estimated that the required capital investment would amount to 84 million KES (840,000 USD) for the purchase of eight exhauster trucks, seven pick-up trucks and one tractor (S1 Table). In addition to these capital investments, ensuring universal access to safe emptying services would require bridging the gap between markets prices and WTP for operational expenses. Assuming that a toilet is emptied on average once per year, we estimated annual costs of 5,000 KES (50 USD) per toilet served by VTOs, and 9,500 KES (95 USD) per toilet served by formal manual emptiers (S1 Table). Serving the additional 8,661 toilets would thus require operational expenditures of 53 million KES (530,000 USD) annually (S1 Table).

Our survey data showed that the first quintile of WTP (the amount that 80% of households stated being willing to pay) was 1,200 KES (12 USD) for VTO services and 2,100 KES (21 USD) for formal manual services. Extrapolating to the 8,661 toilets requiring safe emptying and transport services in Kisumu, we estimated that cumulative WTP would amount to 12.3 million KES (123,000 USD) annually, corresponding to 23% of operational costs (S1 Table). The resulting gap between operational costs and cumulative household WTP would then be 40.7 million KES (407,000 USD) annually, equivalent to 800 KES (8 USD) per unserved household or 4,700 KES (47 USD) per toilet (S1 Table). Estimates in the scenario of a sewer expansion to 20% of unserved households are presented in S1 Table, and result in the same annual financial gap per household or toilet.

## Discussion

### Study summary and key findings

Rapidly growing cities in low-income countries face formidable requirements to provide improved sanitation infrastructure and services that reduce exposures to fecal waste and promote public health and economic development. Strategies to address these sanitation needs must include analyses of financing options, particularly the balance between *market finance* (the amount that householders pay directly or through rents), and *subsidy finance* (some combination of tax-based government funds, tariff-based rich to poor subsidies, and donor inputs) [4]. Information on the amounts that householders are willing to pay directly is critical for these financing analyses. In this study, we measured the amounts that low-income households in the city of Kisumu, Kenya, were willing to pay for safely-managed latrine pit-emptying services. We also compared these amounts to the market prices of these services. We utilized two

methods to determine WTP: household surveys to quantify *stated* WTP and randomized distribution of discount vouchers to quantify *revealed* WTP.

Our surveys of 942 households in low-income areas of Kisumu found that the majority (74%) relied on dry improved pit latrines, and most of the remaining households had pour-flush latrines connected to some form of underground containment (Table 2). FGD respondents noted that fecal sludge emptying was a common necessity in much of Kisumu due to high water tables, and 40% of the surveyed households reported that their underground containment was emptied in the last year (Table 2). The majority (62%) of our surveyed households utilized informal manual pit-emptiers, and most of the remainder reported emptying by a VTO (Table 2). Awareness of the formalized manual emptying businesses established through NGO efforts was low, though households responded favorably when their services were described.

In conjunction with the household surveys, we compared stated and revealed demand for two types of pit-emptying services: manual emptying by the formalized business, Gasia Poa, and vacuum truck emptying. Both stated and revealed WTP were higher for emptying by Gasia Poa (Fig 3A–3C). We surmised that this was because manual emptying was more likely to include the removal of solid waste that is commonly found in pit latrines; in post-voucher distribution follow-up surveys, respondents noted the importance of solid waste removal (Table 3). Additionally, the fact that approximately 25% of survey respondents were not accessible to VTOs likely lowered stated WTP for these services (S2 Table). Revealed WTP was lower than stated WTP for both services, though the differences were only significant for emptying by VTOs (Fig 3B and 3C). In the case of Gasia Poa, we only compared stated and revealed WTP at higher price points, which may have limited our ability to detect significant differences (Fig 3B). Nevertheless, at market prices, stated and revealed demand for both types of emptying services was low–less than 20% of households were willing to pay full market prices (Fig 3B and 3C); these results are comparable to a similar study in Rwanda that found only 15% of pits would be emptied at estimated prices that did not include subsidies [5].

Weak demand for safe emptying services in Kisumu was likely a combination of two factors: low income levels and the prevalence of unregulated, informal manual emptiers. Approximately half of our study population reported monthly household incomes lower than 10,000 KES (100 USD), which other studies estimated as the average income level in Kisumu LIAs [37–39]. Spread over time and amongst households sharing a toilet, safe emptying services represent a modest fraction of incomes (approximately 1%, considering annual costs of 5,000–9,500 KES per toilet shared by six households, S1 Table); despite this, a one-off payment, especially if carried by a single household as is often the case, can represent over 50–100% of a household's monthly income and therefore compete with other priority expenditures. Additionally, informal manual emptiers are currently the most widespread form of service provision in Kisumu LIAs. Because they offer services that are 50–75% cheaper than formal service providers (Table 1), they contribute to keeping demand low. Under these circumstances, we estimated that the gap between household WTP and market prices for providing comprehensive safe emptying services to Kisumu LIAs amounted to 27.6–40.7 million KES (276,000–407,000 USD) annually (S1 Table). This gap may diminish if strict regulations are implemented to eliminate the practice of informal manual emptying.

We estimated annual emptying costs to be 50–95 USD per toilet, or 8–16 USD per household. For comparison, prior studies estimated that the annual costs of safe emptying services, in optimal scenarios with high market penetration, would amount to approximately 4–8 USD per household in Bangladesh [6] and 7 USD per household in Rwanda [5]. Our cost estimates for Kisumu are slightly higher than in these other settings, though extending safe emptying services to all Kisumu households may offer opportunities for economies of scale and reduce

costs. Finally, our emptying cost estimates are notably higher than the assumptions used in global cost projections: in a 2016 report estimating the costs of achieving Sustainable Development Goal 6.2 for universal sanitation, the World Bank estimated that annual OPEX would amount to 1 USD per household for a pit latrine in urban Kenya [40], which is approximately one order of magnitude lower than our estimates based on market prices of safe emptying services in Kisumu.

## Study limitations

This study has several limitations. First, many households were ineligible for the study, the majority (31%) because the financial responsibility of pit emptying fell to landlords that lived offsite. It is likely that landlords living offsite may have a lower WTP for safe emptying services since they would not benefit from health, environmental, or other benefits of those services. Second, it is possible that revealed WTP may have increased if we had given households more time to redeem vouchers; though 10 households in the follow-up survey indicated that their pit was not full enough for emptying, half of these also cited other reasons, and eight met the eligibility criteria of either intending to empty their pit or having visible fecal sludge. Third, we were not able to examine revealed WTP at price points below 3000 KES (30 USD), the lowest price points for Gasia Poa. As described in the methods, we calculated the voucher price points as 20%, 40%, 60%, and 80% of the estimated price of 14,000 KES, reflecting a worst-case scenario with solid waste removal.

## Conclusion

Safely managed onsite sanitation is gaining increasing attention as an alternative to piped sewerage in rapidly expanding cities of the developing world. FSM services, often provided by the private sector, already exist in most cities; expanding their coverage, particularly among lower income communities, is widely perceived as more cost-effective than extending and maintaining piped sewage networks and wastewater treatment plants [41–43]. Sewerage requires large capital investments and massive public infrastructure development. In contrast, scaling-up onsite sanitation is primarily seen as a market-driven activity, largely funded by customers and, in some cases, demand for recycled waste products.

Our results confirm that onsite sanitation facilities are common, particularly among the poor, in a large Kenyan city. They also show, however, that WTP in low-income neighborhoods is too low to support safe emptying services at current market prices. In theory, demand generation activities have the potential to increase WTP; however, previous research has found that sanitation demand interventions for increasing household investments in sanitation are difficult to implement successfully [44, 45]. Cost-reductions have the potential to reduce the gap between WTP and market prices, though we did not find opportunities for substantial cost-savings among pit-emptying services in Kisumu [36]. Finally, emerging technologies may reduce the costs of safe sanitation provision: for example, the waste recovery and reuse components of CBS can offset treatment costs [46]; and the application of selected worm species to promote fecal sludge decomposition in pit latrines (vermifiltration) can reduce emptying requirements [21]. Nevertheless, improving FSM in these areas will likely require large and on-going subsidies. Beyond subsidies to support operating costs, service providers will need to invest in emptying and transportation equipment to serve larger populations, and the city will need additional fecal sludge treatment capacity.

Addressing these needs for subsidies and capital investments will entail substantial coordination, planning and investment by government and donor agencies. The complexity, if not the expense, of implementation may compare to traditional sewerage development.

Furthermore, it is unlikely that recently created and legally operated manual emptying businesses will compete effectively with lower-cost illegal operations. Providing safe and affordable sanitation for all residents of Kisumu, therefore, will necessitate a comprehensive sanitation market plan for the city that includes programs for subsidizing FSM for the poor, investing in transport and treatment of fecal sludge, coordination with expansion of sewerage networks, and enforcement of FSM regulations. In the absence of master plan development and execution, it is unlikely that small and fragmented FSM efforts will achieve substantial impacts, particularly among the poor.

## Supporting information

**S1 Fig.** Graphics of (a) formal manual emptying, and (b) formal manual disposal, and (c) VTOs. We used to explain these services to study participants.
(TIF)

**S2 Fig. Example Gasia Poa voucher.** The randomized starting price points were: 3,000/6,000/8,000/11,000 KES (30/60/80/110 USD) for Gasia Poa emptying services. The full cost for Gasia Poa emptying services was 14,000 KES (140 USD).
(TIF)

**S1 Table. Financial analysis of the capital expenditures (CAPEX) and operating expenditures (OPEX) for safe emptying in Kisumu, Kenya and comparisons with customer willingness-to-pay.**
(DOCX)

**S2 Table. Bivariate subgroup analysis of stated WTP.**
(DOCX)

**S3 Table. Multivariate subgroup analysis of stated WTP.**
(DOCX)

**S4 Table. Bivariate and multivariate subgroup analysis of voucher redemption.**
(DOCX)

**S1 Text. Household survey in English.**
(DOCX)

**S2 Text. Household survey in Luo.**
(DOCX)

## Acknowledgments

We appreciate the contributions from the Water & Sanitation for the Urban Poor (WSUP) team including Guy Norman, Sam Drabble, Rosie Renouf, and Emanuel Owako. We gratefully acknowledge our enumerators and enumerator supervisor Denish Owiti for their contributions. We also specifically thank the following Aquaya staff: Joyce Kisiangani for her contributions to enumerator training, Brian Mwangi for his contributions to enumerator training and tracking voucher redemptions, Dianne Vernon for her contributions to fieldwork management, and Sale Rhodes for her contributions to data analysis. We also thank all of the interviewees, FGD participants, and households that contributed to this study.

## Author Contributions

**Conceptualization:** Rachel Peletz, Caroline Delaire, Ranjiv Khush.

**Data curation:** Alicea Easthope-Frazer, Caroline Delaire.

**Formal analysis:** Rachel Peletz.

**Funding acquisition:** Ranjiv Khush.

**Methodology:** Rachel Peletz, Alicea Easthope-Frazer, Caroline Delaire, Ranjiv Khush.

**Project administration:** Clara MacLeod, Joan Kones, Edinah Samuel.

**Supervision:** Caroline Delaire.

**Writing – original draft:** Rachel Peletz, Clara MacLeod, Ranjiv Khush.

**Writing – review & editing:** Rachel Peletz, Clara MacLeod, Caroline Delaire, Ranjiv Khush.

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
