## [Decision Letter · Decision Letter 0]

29 May 2020

PONE-D-20-07368

When Pits Fill Up: Supply and Demand for Safe Pit-Emptying Services in Kisumu, Kenya

PLOS ONE

Dear Dr. Khush,

Thank you for submitting your manuscript to PLOS ONE. After careful consideration, we feel that it has merit but does not fully meet PLOS ONE’s publication criteria as it currently stands. Therefore, we invite you to submit a revised version of the manuscript that addresses the points raised during the review process.

This is an article that follows STROBE guidelines for observational studies. Please follow the submission guidelines in presenting the whole text. Minor revisions as pointed out reviewers should be considered to strengthen scientific integrity.

We look forward to receiving your revised manuscript.

Kind regards,

Khin Thet Wai, MBBS, MPH, MA (Population & Family Planning Research)

Academic Editor

PLOS ONE

Journal Requirements:

2. Please include additional information regarding the survey or questionnaire used in the study and ensure that you have provided sufficient details that others could replicate the analyses.

For instance, if you developed a questionnaire as part of this study and it is not under a copyright more restrictive than CC-BY, please include a copy, in both the original language and English, as Supporting Information.

5. We note that Supplementary Figure 1 in your submission contain map images which may be copyrighted.

5.1.    You may seek permission from the original copyright holder of Supplementary Figure 1 to publish the content specifically under the CC BY 4.0 license.

5.2.    If you are unable to obtain permission from the original copyright holder to publish this figure under the CC BY 4.0 license or if the copyright holder’s requirements are incompatible with the CC BY 4.0 license, please either i) remove the figure or ii) supply a replacement figure that complies with the CC BY 4.0 license. Please check copyright information on all replacement figures and update the figure caption with source information. If applicable, please specify in the figure caption text when a figure is similar but not identical to the original image and is therefore for illustrative purposes only.

Reviewers' comments:

Reviewer's Responses to Questions

**Comments to the Author**

1. Is the manuscript technically sound, and do the data support the conclusions?

Reviewer #1: Yes

Reviewer #2: Yes

2. Has the statistical analysis been performed appropriately and rigorously? 

Reviewer #1: Yes

Reviewer #2: Yes

3. Have the authors made all data underlying the findings in their manuscript fully available?

Reviewer #1: Yes

Reviewer #2: No

4. Is the manuscript presented in an intelligible fashion and written in standard English?

Reviewer #1: Yes

Reviewer #2: Yes

5. Review Comments to the Author

Reviewer #1: This study represents an important contribution to the sanitation research literature and I recommend publication after the following two suggestions for improvement have been taken into account:

1. The authors briefly mention contained-based sanitation (CBS) models in the discussion part of their paper and correctly state that these models may represent a solution to bridge the gap between faecal sludge management costs and willingness-to-pay. However, even in locations where latrines and septic tanks are already in place, increasing efforts to recover value from the sludge in the way that is done in CBS models (e.g. recovering biogas or making soil amendment out of the sludge) has an important role to play in reducing the financial gap also. I feel these points (CBS, value recovery from sludge) should be emphasised more strongly throughout the paper (e.g. in the abstract and conclusions) rather than concluding that government subsidies are the only solution for filling the financial gap.

2. Another solution to this problem would be to reduce the frequency of pit emptying, e.g. by promoting the installation of vermifiltration toilets instead of conventional latrines / septic tanks. These toilets fill up much less quickly, and when they do fill up, their contents are relatively sterile compost material which is safe to empty and dispose of, instead of hazardous faecal sludge. See: Furlong et al, 2015, The development of an onsite sanitation system based on vermifiltration: the 'Tiger Toilet', Journal of Water, Sanitation and Hygiene for Development, Vol:5, 608-613.

Reviewer #2: This is a good paper that I enjoyed reading, and only needs minor changes before accepting. I think it sets new standards methodologically for WTP studies of emptying services.

Main points:

1. The intro would merit some indication of a gap in the literature. The authors note a couple of other WTP studies for emptying services but it would be good to specifically argue (in intro or in discussion) what this paper specifically adds to that. How do you methods differ from what’s been done before and do you perceive them as “better”, and why?

2. Study site / setting section needs to be expanded (mostly a case of moving results material up). I know the authors see their interview findings are results but it is better to put some of that info up front. E.g. extent of tenancy, extent of sharing, explaining “compounds”, an idea of extent of wet v dry pit latrines within that big 70% (which has big implications for emptying tech/price as you note later). There’s no point leaving this until results as it helps justify methods. The fact that we don’t find out until p.14 that sharing is v common (85% in your sample) is problematic. How does this compare to city as a whole?

3. It was a shame informal manual emptying wasn’t included in the stated pref component. Obviously vouchers for them would have been unworkable but at least you could then have compared demand curve for manual emptying to your two services. They are the market leader. I am sure you had your reasons, but at least set out what they are (potentially in discussion, or in methods)

4. When you’re talking about your WTP variable, be clear about whether it’s maximum WTP, median, mean etc. – sometimes this is done and sometimes not. Likewise costs, be specific when you’re talking about financial cost from the provider perspective. Also “price” which is used a bit loosely (“controlling for price”). You know which cost/price you’re talking about but the reader doesn’t, so be specific.

5. Tenant WTP for rent increases as a way to value services is an interesting idea. If there are other studies that did this in other sectors you should cite them. If it hasn’t been done before, then say so.

6. Rational / precedent for 80% of HHs WTP as the gap cut-off? Likewise no allowance for profit or capital recovery in your price = OpEx assumption?

7. In table 1 you put range of 9k-12k (price) for formal manual SPs emptying a pit latrine, but higher up you said 14k for Gasia Poa (cost). Are you implying GP is subsidised? Likewise re table 1 the market share data presented in results would just make sense to be introduced here. I know it’s technically a result but it interrupts the flow for someone (like me) who doesn’t know Kisumu.

8. The demand curves are beautiful. To what extent is lower WTP for VTOs biased by accessibility, e.g. HHs far from roads saying "Of course I'm not WTP anything for VTO because they can't get here". I think this was accounted for by your design for RP, but possibly not skipping patterns for SP. It seems problematic if not. If so, consider a sub-group analysis on SP for VTOs for people living <50m from road? Line 403 – are these 95%CIs? if yes say so. If not, why are there ranges.

9. Financial requirements – you talk about number of HHs but, given the huge extent of sharing, number of toilets may be more appropriate. you bring this in later but better to be upfront. I get that it’s modelled in the supp mats but need to be clear in main body.

10. Line 495 - annual emptying costs of 100 USD per toilet for VTOs, 190 USD by formal manual emptiers. These are huge numbers. It’s kind of implicit in the discussion – you’d do better to emphasise just how much that is. it’s not much cheaper than building a new toilet every year. Consider that in Hutton & Varughese’s 2016 SDG costing, urban pit latrines are assumed to have ~5% annual OpEx, and 30% CapManEx every 4 years. Your estimates are several orders of magnitude higher. Proper FSM services are expensive. you make the point in the discussion that there’s not that much more room for driving costs down, but you could more clearly articulate just how expensive these services are relative to what people think (incl. SDG costing). You must have latrine construction data for Kisumu so express annual emptying costs as % of typical annualised CapEx, or some other metric?

11. Line 510 – I got a bit lost by the end of the calculations. Is there a way to make it more visual?

12. Line 529 – “40% of the surveyed households reported that their underground containment was emptied in the last year”. But in your model (line 503) you say “Assuming that the average toilet is emptied every six months”. Unless I'm missing something, that’s appears a little bit inconsistent… so explain why.

13. Line 546 – “This weak demand can also be interpreted as a general constraint of low income” – is it not more explained by the market being dominated by cheap manual emptiers? Without actually enforcing a ban, or otherwise increasing the true cost of using them, then it will remain the issue. You don’t address this.

14. “Expanding safe emptying for the poor” – this section feels odd – a summary of what’s going on in the sector. Kind of not necessary in this depth – you can engage with your own findings a lot more, or integrate better with rest of discussion

6. PLOS authors have the option to publish the peer review history of their article (what does this mean?). If published, this will include your full peer review and any attached files.

Reviewer #1: No

Reviewer #2: Yes: Ian Ross

---

## [Author Response · Author response to Decision Letter 0]

28 Jul 2020

Dear Director Khin Thet Wai and Editors of Plos One,

We thank you for the opportunity to respond to the Reviewers’ comments, which have helped improve the clarity of our manuscript. In this response document, we have indicated the Reviewers’ comments in black, our answers in blue, and resulting changes to the manuscript in red. We prepared two revised manuscripts: with and without “track changes.” The line numbers indicated in this response document correspond to the revised manuscript with track changes.

Comments by Reviewer #1

This study represents an important contribution to the sanitation research literature and I recommend publication after the following two suggestions for improvement have been taken into account:

1. The authors briefly mention contained-based sanitation (CBS) models in the discussion part of their paper and correctly state that these models may represent a solution to bridge the gap between faecal sludge management costs and willingness-to-pay. However, even in locations where latrines and septic tanks are already in place, increasing efforts to recover value from the sludge in the way that is done in CBS models (e.g. recovering biogas or making soil amendment out of the sludge) has an important role to play in reducing the financial gap also. I feel these points (CBS, value recovery from sludge) should be emphasised more strongly throughout the paper (e.g. in the abstract and conclusions) rather than concluding that government subsidies are the only solution for filling the financial gap.

2. Another solution to this problem would be to reduce the frequency of pit emptying, e.g. by promoting the installation of vermifiltration toilets instead of conventional latrines / septic tanks. These toilets fill up much less quickly, and when they do fill up, their contents are relatively sterile compost material which is safe to empty and dispose of, instead of hazardous faecal sludge. See: Furlong et al, 2015, The development of an onsite sanitation system based on vermifiltration: the 'Tiger Toilet', Journal of Water, Sanitation and Hygiene for Development, Vol:5, 608-613.

Response to both comments: We thank the reviewer for highlighting both waste reuse for improving the reach and sustainability of sanitation services and the emergence of new technologies. We have noted these points in the conclusion (Lines 1017-1021):

Finally, emerging technologies may reduce the costs of safe sanitation provision: for example, the waste recovery and reuse components of CBS can offset treatment costs (46); and the application of selected worm species to promote fecal sludge decomposition in pit latrines (vermifiltration) can reduce emptying requirements (21).

Comments by Reviewer #2

This is a good paper that I enjoyed reading, and only needs minor changes before accepting. I think it sets new standards methodologically for WTP studies of emptying services.

We thank the reviewer for the positive feedback and constructive comments. Our responses are noted below.

Main points:

1. The intro would merit some indication of a gap in the literature. The authors note a couple of other WTP studies for emptying services but it would be good to specifically argue (in intro or in discussion) what this paper specifically adds to that. How do you methods differ from what’s been done before and do you perceive them as “better”, and why?

Response: We have added the following study justification to the introduction, lines 99-109:

To support sanitation improvements in Kisumu, the third largest city of Kenya, we measured the gap between market prices and WTP for safe emptying services in the city’s low-income areas. Stated and revealed willingness-to-pay (WTP) methods provide options for quantifying demand for improved sanitation products and services among low-income households. Stated WTP methods generally use household surveys to assess demand for hypothetical products and services by asking respondents to reflect on price points through a series of questions (Breidert et al., 2006; Cummings et al., 2016; Mitchell and Carson, 1989; Whittington et al., 1990). In contrast, revealed WTP methods obtain results from real money price-responses (i.e., market data or experiments) (Breidert et al., 2006) and are generally considered more reliable because they reflect respondents’ actual (rather than hypothetical) purchasing behaviors (Breidert et al., 2006). Our study, therefore, addressed the following objectives: 1) to compare household demand for safe emptying services with the prices charged for these services; and 2) to compare stated and revealed WTP by conducting both household surveys and a real-money voucher experiment. 

2. Study site / setting section needs to be expanded (mostly a case of moving results material up). I know the authors see their interview findings are results but it is better to put some of that info up front. E.g. extent of tenancy, extent of sharing, explaining “compounds”, an idea of extent of wet v dry pit latrines within that big 70% (which has big implications for emptying tech/price as you note later). There’s no point leaving this until results as it helps justify methods. The fact that we don’t find out until p.14 that sharing is v common (85% in your sample) is problematic. How does this compare to city as a whole?

Response: We have moved the information on pit-emptying practices and services, and our study population to the Methods as sub-sections under “Market Assessment”. We have clarified that approximately 30% of households are estimated to share toilets across Kisumu as a whole. 

3. It was a shame informal manual emptying wasn’t included in the stated pref component. Obviously vouchers for them would have been unworkable but at least you could then have compared demand curve for manual emptying to your two services. They are the market leader. I am sure you had your reasons, but at least set out what they are (potentially in discussion, or in methods)

Response: Our main reason for not measuring stated WTP for informal manual emptying was that the market had already established prices of 1,000-3,000 KES for our study households. We have clarified this point in the Methods section, lines 315-316:

We did not measure WTP for informal manual emptying because the price that study households paid for informal emptying was well established at 1,000 – 3,000 KES (Table 2). 

In addition, our survey included WTP questions for other sanitation products and services (manuscripts in preparation), and we were concerned about overwhelming respondents and compromising our data quality. 

4. When you’re talking about your WTP variable, be clear about whether it’s maximum WTP, median, mean etc. – sometimes this is done and sometimes not. Likewise costs, be specific when you’re talking about financial cost from the provider perspective. Also “price” which is used a bit loosely (“controlling for price”). You know which cost/price you’re talking about but the reader doesn’t, so be specific.

Response: We have clarified the WTP variable in several locations throughout the manuscript. We have also amended the Methods as indicated below. We also want to point out that measuring WTP through voucher redemption does not allow calculating a mean, median or maximum WTP, which is why we simply refer to “revealed WTP” in the Results. 

Lines 347-350: We followed the dichotomous choice questions with an open-ended question asking respondents to state their WTP amount the highest amount they would be willing to pay (Hanemann et al., 1991). We report this maximum WTP as “stated WTP” in the rest of the paper. 

We thank the Reviewer for pointing out the loose usage of price vs. costs. We have rectified the manuscript in several locations to distinguish market prices from costs to service providers. (We initially used the phrase “full costs” to signify “market prices”).

5. Tenant WTP for rent increases as a way to value services is an interesting idea. If there are other studies that did this in other sectors you should cite them. If it hasn’t been done before, then say so.

Response: Two prior studies using hedonic pricing methods in Kisumu, Kenya (Simiyu, Swilling, Rheingans, & Cairncross, 2017), and Lusaka, Zambia (Tidwell, Terris-prestholt, Quaife, & Aunger, 2019), demonstrated that tenants’ willingness-to-pay for sanitation can be quantified by comparing rents of dwellings with and without these facilities. Similarly, a study in Guatamela documented how demand for utility services such as water, sanitation, electricity, and telephone is reflected in rent premia (Foster & Araujo, 2004). We have added these references to the manuscript as justification of our methods:

Lines 369-373: Prior studies have demonstrated that tenants’ demand for water and sanitation services is reflected in rent premia (Foster & Araujo, 2004; Simiyu et al., 2017; Tidwell et al., 2019). Therefore, when surveying tenants who were not financially responsible for emptying services, we asked about their WTP for an increase in rent (rather than WTP for one-time outlay or subscription services) if emptying services were provided when necessary in their dwelling.

6. Rational / precedent for 80% of HHs WTP as the gap cut-off? Likewise no allowance for profit or capital recovery in your price = OpEx assumption?

Response: To estimate household subsidy requirements, we wanted to choose a cut-off price point that the majority of households were willing to pay; this is why we chose a proportion substantially larger than 50% (which would have corresponded to the median WTP). The specific proportion of 80% was however arbitrary. The corresponding price points (12 USD for VTOs and 21 USD for formal manual emptying) were consistent with what LIA residents currently pay for informal manual emptying services (10-30 USD, Table 1). As a result, we did not feel that choosing a larger proportion (corresponding to lower price points) was justified. We modified the manuscript as follows:

Lines 487-489: To calculate the gaps between service delivery costs and customer WTP, we selected that amounts To estimate household subsidy requirements, we compared OPEX to the amount that approximately 80% of households were willing to pay for safe emptying services (S1 Table). We verified that this amount was consistent with current market prices for informal manual emptying (Table 1). 

Regarding the second point raised by the Reviewer: our OPEX estimates do account for profit and capital depreciation since we have equated OPEX to market prices. We have rectified the manuscript as follows:

Lines 485-487: We approximated assumed that operational expenditures (OPEX) were entirely captured in the existing average consumer using average market prices, which capture capital depreciation and possible service provider margins.

7. In table 1 you put range of 9k-12k (price) for formal manual SPs emptying a pit latrine, but higher up you said 14k for Gasia Poa (cost). Are you implying GP is subsidised? Likewise re table 1 the market share data presented in results would just make sense to be introduced here. I know it’s technically a result but it interrupts the flow for someone (like me) who doesn’t know Kisumu.

Response: Table 1 provides the typical market prices exclusive of solid waste removal. In contrast, the price of 14,000 KES was inclusive of solid waste removal. We have rectified the manuscript as follows:

Table 1: These groups charge an additional flat fee for solid waste removal from pit latrines.

Lines 353 and 412: we estimated the full costs market prices to be 14,000 KES (140 USD) for Gasia Poa (inclusive of an additional fee for solid waste removal).

Line 887-1001: As described in the methods, we calculated the voucher price points as 20%, 40%, 60%, and 80% of the estimated price of 14,000 KES however, this initial estimate was higher than common market prices (7,000-12,000 KES), reflecting a worst-case scenario with solid waste removal.

8. The demand curves are beautiful. To what extent is lower WTP for VTOs biased by accessibility, e.g. HHs far from roads saying "Of course I'm not WTP anything for VTO because they can't get here". I think this was accounted for by your design for RP, but possibly not skipping patterns for SP. It seems problematic if not. If so, consider a sub-group analysis on SP for VTOs for people living <50m from road? Line 403 – are these 95%CIs? if yes say so. If not, why are there ranges.

Response: We thank the Reviewer for pointing out this possible bias. We indeed found that WTP for VTO services was higher amongst households who had a toilet less than 50 m from a road (median of 2,500 KES) compared to households who didn’t (median of 1,000 KES). We have added this insight in Table S2, in the Results, and in the Discussion:

Line 631: Survey respondents had a significantly higher stated WTP for VTO services if they were wealthier (p<0.01), were tenants (p<0.01), and had a toilet located less than 50 m from a road (p<0.01) (Tables S2 and S3).

Line 843: Additionally, the fact that approximately 25% of survey respondents were not accessible to VTOs likely lowered stated WTP for these services (Table S2).

Regarding the Reviewer’s second point: 95% confidence intervals are reported in Figure 3, but this is not what this specific section refers to. We have rearranged the paragraph for improved clarity: 

Line 503: Median stated WTP was 3,000 KES (30 USD) for Gasia Poa services (mean=3,438 KES; 34 USD), representing approximately 25-43% of market prices (which ranged from 7,000-12,000 KES, 70-120 USD). Less than 20% of respondents were willing to pay the lower-bound market price of 7,000 KES (70 USD) (Fig 3a). Median stated WTP was 2,000 KES (20 USD) for VTO services (mean=2,182 KES, 22 USD), representing approximately 33-50% of market prices (which ranged from 4,000-6,000 KES, 40-60 USD). 14% of respondents had a stated WTP of 4,000 KES (40 USD), the lowest market price for VTO services (Fig 3a). Median stated WTP was approximately 33-50% of full costs (4,000-6,000 KES, 40-60 USD) for VTO services and approximately 25-43% of full costs (7,000-12,000 KES, 70-140 USD) for Gasia Poa services.

9. Financial requirements – you talk about number of HHs but, given the huge extent of sharing, number of toilets may be more appropriate. you bring this in later but better to be upfront. I get that it’s modelled in the supp mats but need to be clear in main body.

Response: We have rectified the manuscript to more explicitly present results in terms of toilets requiring safe emptying services (as opposed to households). See specific text revisions in the response to comment #11.

10. Line 495 - annual emptying costs of 100 USD per toilet for VTOs, 190 USD by formal manual emptiers. These are huge numbers. It’s kind of implicit in the discussion – you’d do better to emphasise just how much that is. it’s not much cheaper than building a new toilet every year. Consider that in Hutton & Varughese’s 2016 SDG costing, urban pit latrines are assumed to have ~5% annual OpEx, and 30% CapManEx every 4 years. Your estimates are several orders of magnitude higher. Proper FSM services are expensive. you make the point in the discussion that there’s not that much more room for driving costs down, but you could more clearly articulate just how expensive these services are relative to what people think (incl. SDG costing). You must have latrine construction data for Kisumu so express annual emptying costs as % of typical annualised CapEx, or some other metric?

Response: The Reviewer brings up an important consideration. First, in response to comment #12, we have revised our emptying cost estimates down (by changing the emptying frequency from 6 months to 12 months). The new estimates are 50 USD/toilet/year for VTO services and 95 USD/toilet/year for formal manual services. These correspond to 8-16 USD/household/year, given that a toilet in Kisumu LIAs serves a median of 6 households. Here are a few comparison points:

- The average household income in Kisumu LIAs is approximately 100 USD/month, or 1200 USD/year (Simiyu, Cairncross, & Swilling, 2019; USAID & Concern Worldwide, 2013, 2014). Our estimates of emptying costs thus represent approximately 1% of household income. We have added this point in the Discussion; see edits relative to comment #13.

- In a separate publication, we report our estimates of CAPEX for a pour-flush toilet connected to a lined pit in Kisumu: 908 USD (inclusive of a high-quality superstructure addressing safety, privacy and inclusion) (Delaire et al., n.d.). In this context, annual emptying costs represent 6-10% of CAPEX.

- Hutton & Varughese 2016 estimate total OPEX at 5% of CAPEX, which for a wet pit latrine in urban Kenya equates to: 5% * CAPEX of 44 USD/person (Table E.1) * 5 persons/households / lifetime of 8 years = 1 USD/household/year.

We have added a section in the Discussion to compare our estimates with existing literature (including Hutton & Varughese 2016):

Line 867: We estimated annual emptying costs to be 50-95 USD per toilet, or 8-16 USD per household. For comparison, prior studies estimated that the annual costs of safe emptying services, in optimal scenarios with high market penetration, would amount to approximately 4-8 USD per household in Bangladesh (Balasubramanya et al., 2017) and 7 USD per household in Rwanda (Burt, Sklar, & Murray, 2019). Our cost estimates for Kisumu are slightly higher than in these other settings, though extending safe emptying services to all Kisumu households may offer opportunities for economies of scale and reduce costs. Finally, our emptying cost estimates are notably higher than the assumptions used in global cost projections: in a 2016 report estimating the costs of achieving Sustainable Development Goal 6.2 for universal sanitation, the World Bank estimated that annual OPEX would amount to 1 USD per household for a pit latrine in urban Kenya (Hutton & Varughese, 2016), which is approximately one order of magnitude lower than our estimates based on market prices of safe emptying services in Kisumu. 

11. Line 510 – I got a bit lost by the end of the calculations. Is there a way to make it more visual?

Response: To minimize confusion, we have reduced the content of this section to only present results for the no-sewerage scenario. The results for the sewerage scenario remain in Table S1, but do not add more information since the financial requirement per household remains identical in both scenarios.

Old version: To expand formalized fecal sludge emptying and transportation services to the approximately 51,965 households of Kisumu that are currently underserved we estimated that the required capital investment would amount to 84 million KES (840,000 USD) for the purchase of eight exhauster trucks, seven pick-up trucks and one tractor (Table S1). In the scenario of a sewer expansion to 20% of households, the capital investment required to serve the remaining 35,202 households would amount to 44.5 million KES (445,000 USD) for the purchase four exhauster trucks, four pick-up trucks and one tractor (Table S1). We have not included these capital costs in the following estimates of the variance between operating costs and customer demand.

Based on an average of two emptyings per year, we estimated annual costs of 10,000 KES (100 USD) per toilet served by VTOs, and 19,000 KES (190 USD) per toilet served by formal manual emptiers (Table S1). Assuming that the average toilet is shared by six households, serving the additional 35,202-51,965 households (or 5,867-8,661 toilets) would thus require between 71.9 million KES (719,000 USD) and 106 million KES (1,061,000 USD) annually, depending on whether or not sewer services are expanded (Table S1). 

Our demand data showed that approximately 80% of households were willing to pay 1,200, KES (12 USD) for emptying by a VTO and 2,100 KES (21 USD) for emptying by formal manual emptiers. Assuming that the average toilet is emptied every six months (Table 2), annual WTP thus amounts to 2,400 KES (24 USD) per toilet served for VTOs, and 4,200 KES (42 USD) per toilet served for formal manual emptiers (Table S1). Extrapolating to the 5,867-8,661 toilets requiring safe emptying and transport services in Kisumu, we estimated that cumulative WTP would range from 16.7 million KES (167,000 USD) to 24.7 million KES (247,000 USD) annually, depending on whether or not sewer services are expanded (Table S1); WTP therefore translates to 23% of operational costs. The resulting variances between operational costs and cumulative household WTP would then range from 55.1-81.4 million KES (551,000-814,000 USD) annually, equivalent to 1600 KES (16 USD) per unserved household (Table S1).

New version, line 688: To expand formalized fecal sludge emptying and transportation services to the approximately 51,965 households of Kisumu that are currently underserved (corresponding to approximately 8,661 toilets), we estimated that the required capital investment would amount to 84 million KES (840,000 USD) for the purchase of eight exhauster trucks, seven pick-up trucks and one tractor (Table S1). In addition to these capital investments, ensuring universal access to safe emptying services would require bridging the gap between markets prices and WTP for operational expenses. Assuming that a toilet is emptied on average once per year, we estimated annual costs of 5,000 KES (50 USD) per toilet served by VTOs, and 9,500 KES (95 USD) per toilet served by formal manual emptiers (Table S1). Serving the additional 8,661 toilets would thus require operational expenditures of 53 million KES (530,000 USD) annually (Table S1). 

Our survey data showed that the first quintile of WTP (the amount that 80% of households stated being willing to pay) was 1,200 KES (12 USD) for VTO services and 2,100 KES (21 USD) for formal manual services. Extrapolating to the 8,661 toilets requiring safe emptying and transport services in Kisumu, we estimated that cumulative WTP would amount to 12.3 million KES (123,000 USD) annually, corresponding to 23% of operational costs (Table S1). The resulting gap between operational costs and cumulative household WTP would then be 40.7 million KES (407,000 USD) annually, equivalent to 800 KES (8 USD) per unserved household or 4,700 KES (47 USD) per toilet (Table S1). Estimates in the scenario of a sewer expansion to 20% of unserved households are presented in Table S1, and result in the same annual financial gap per household or toilet.

12. Line 529 – “40% of the surveyed households reported that their underground containment was emptied in the last year”. But in your model (line 503) you say “Assuming that the average toilet is emptied every six months”. Unless I'm missing something, that’s appears a little bit inconsistent… so explain why.

Response: Our data unfortunately did not allow us to precisely estimate the emptying frequency (we asked respondents about the time since the last emptying, as opposed to the emptying frequency, because we felt that the former would be more accurate). The Reviewer is however correct that the frequency of six-months was not very consistent with our survey data. We have therefore rectified our calculations using an emptying frequency of 12 months (instead of 6 months) and justified this decision in Table S1:

Table S1: Assumption based on survey data. Among voucher recipients (who all stated planning to empty their pit in the next three months), approximately half had their pit last emptied less than one year ago, and approximately half had it last emptied more than one year ago or never (Table 2). We therefore used one year as a plausible average frequency of emptying in Kisumu. 

The new financial estimates are reflected in our response to comment #11.

13. Line 546 – “This weak demand can also be interpreted as a general constraint of low income” – is it not more explained by the market being dominated by cheap manual emptiers? Without actually enforcing a ban, or otherwise increasing the true cost of using them, then it will remain the issue. You don’t address this.

Response: the reviewer has noted an important point, which has motivated us to clarify our presentation of emptying costs vs income. We have provided edits as follow:

Line 852: Weak demand for safe emptying services in Kisumu was likely a combination of two factors: low income levels and the prevalence of unregulated, informal manual emptiers. Approximately half of our study population reported monthly household incomes lower than 10,000 KES (100 USD), which other studies estimated as the average income level in Kisumu LIAs (Simiyu et al., 2019; USAID & Concern Worldwide, 2013, 2014). Spread over time and amongst households sharing a toilet, safe emptying services represent a modest fraction of incomes (approximately 1%, considering annual costs of 5,000-9,500 KES per toilet shared by six households, Table S1); despite this, a one-off payment, especially if carried by a single household as is often the case, can represent over 50-100% of a household’s monthly income and therefore compete with other priority expenditures. Additionally, informal manual emptiers are currently the most widespread form of service provision in Kisumu LIAs. Because they offer services that are 50-75% cheaper than formal service providers (Table 1), they contribute to keeping demand low. Under these circumstances, we estimated that the gap between household WTP and market prices for providing comprehensive safe emptying services to Kisumu LIAs amounted to 27.6-40.7 million KES (276,000-407,000 USD) annually (Table S1). This gap may diminish if strict regulations are implemented to eliminate the practice of informal manual emptying.

14. “Expanding safe emptying for the poor” – this section feels odd – a summary of what’s going on in the sector. Kind of not necessary in this depth – you can engage with your own findings a lot more, or integrate better with rest of discussion

Response: We have moved most of the material in the previous “Expanding Safe Emptying for the Poor” section to the introduction to provide more context regarding urban sanitation interventions.

Comments by the Editor

Response: We have rectified the manuscript style as required.

2. Please include additional information regarding the survey or questionnaire used in the study and ensure that you have provided sufficient details that others could replicate the analyses.

For instance, if you developed a questionnaire as part of this study and it is not under a copyright more restrictive than CC-BY, please include a copy, in both the original language and English, as Supporting Information.

Response: We have added the survey questionnaire in Luo and English in the Supporting Information.

Response: We confirm that we will make the data publicly available via Mendeley Data upon manuscript acceptance. 

Response: We have made the requested change. 

5. We note that Supplementary Figure 1 in your submission contain map images which may be copyrighted.

Response: We have deleted the previous Supplementary Figure S1. The authors generated the current supplementary Figures S1 and S2.

---

## [Decision Letter · Decision Letter 1]

7 Aug 2020

When Pits Fill Up: Supply and Demand for Safe Pit-Emptying Services in Kisumu, Kenya

PONE-D-20-07368R1

Dear Dr. Khush,

We’re pleased to inform you that your manuscript has been judged scientifically suitable for publication and will be formally accepted for publication once it meets all outstanding technical requirements.

Kind regards,

Khin Thet Wai, MBBS, MPH, MA (Population & Family Planning Resear

Academic Editor

PLOS ONE

Additional Editor Comments (optional):

Reviewers' comments:

Reviewer's Responses to Questions

**Comments to the Author**

1. If the authors have adequately addressed your comments raised in a previous round of review and you feel that this manuscript is now acceptable for publication, you may indicate that here to bypass the “Comments to the Author” section, enter your conflict of interest statement in the “Confidential to Editor” section, and submit your "Accept" recommendation.

Reviewer #2: All comments have been addressed

2. Is the manuscript technically sound, and do the data support the conclusions?

Reviewer #2: Yes

3. Has the statistical analysis been performed appropriately and rigorously? 

Reviewer #2: Yes

4. Have the authors made all data underlying the findings in their manuscript fully available?

Reviewer #2: Yes

5. Is the manuscript presented in an intelligible fashion and written in standard English?

Reviewer #2: Yes

6. Review Comments to the Author

Reviewer #2: All revisions are clear and make sense. I have no further suggestions to make. My answer to "Have the authors made all data underlying the findings in their manuscript fully available" is based on authors' response to the editor in their letter.

7. PLOS authors have the option to publish the peer review history of their article (what does this mean?). If published, this will include your full peer review and any attached files.

Reviewer #2: **Yes: **Ian Ross

---

## [Editor Report · Acceptance letter]

21 Aug 2020

PONE-D-20-07368R1 

When Pits Fill Up: Supply and Demand for Safe Pit-Emptying Services in Kisumu, Kenya 

Dear Dr. Khush:

I'm pleased to inform you that your manuscript has been deemed suitable for publication in PLOS ONE. Congratulations! Your manuscript is now with our production department. 

Kind regards, 

on behalf of

Dr. Khin Thet Wai 

Academic Editor

PLOS ONE